# Psychometric Properties of the Spanish Version of the Fatigue Assessment Scale in Caregivers of Palliative Care Patients

**DOI:** 10.3390/jcm11143999

**Published:** 2022-07-11

**Authors:** Ana A. Esteban-Burgos, Manuel Fernández-Alcántara, Silvia Escribano, Juana Perpiñá-Galvañ, Concepción Petra Campos-Calderón, María José Cabañero-Martínez

**Affiliations:** 1Department of Nursing, Faculty of Health Sciences, Instituto de Investigación Biosanitaria (ibs.GRANADA), University of Granada, 18016 Granada, Spain; anaestebanburgos@ugr.es; 2Department of Health Psychology, Campus de San Vicente del Raspeig, University of Alicante, San Vicente de el Raspeig, 03690 Alicante, Spain; 3Department of Nursing, Faculty of Health Sciences, Institute for Health and Biomedical Research (ISABIAL), University of Alicante, San Vicente del Raspeig, 03690 Alicante, Spain; silvia.escribano@ua.es (S.E.); juana.perpina@ua.es (J.P.-G.); mariajose.cabanero@ua.es (M.J.C.-M.); 4Servicio Andaluz de Salud (SAS), 41071 Seville, Spain; concha_campos@hotmail.com

**Keywords:** fatigue, fatigue assessment scale, palliative care, caregivers, carers

## Abstract

Symptoms of fatigue and lack of energy are very common in caregivers of palliative care (PC) patients, traditionally associated with variables such as burden or depression. There are no Spanish-language instruments validated for assessing fatigue levels in this population. The Fatigue Assessment Scale (FAS) is a useful and simple instrument for assessing fatigue in this group. The aim of this study was to examine its psychometric properties (factor structure, reliability and validity) in a sample of caregivers of PC patients. Instrumental design for instrument validation was performed. One hundred and eight caregivers of PC patients participated and completed measures of fatigue, family functioning, life satisfaction, caregiver burden, anxiety, depression, resilience and quality of life. A confirmatory factor analysis was performed; non-linear reliability coefficient and Pearson correlations and t-tests were conducted to assess evidence of reliability and validity. The Spanish version of the FAS was found to have a one-dimensional structure. Reliability was 0.88. Validity evidence showed that FAS scores were positively associated with levels of burden, anxiety and depression. They were negatively associated with family functioning, life satisfaction, resilience and quality of life. The Spanish version of the FAS in caregivers of PC patients shows adequate psychometric properties.

## 1. Introduction

Fatigue is a non-specific condition for which there is no precise definition, even though its prevalence is high in both healthy people and those suffering from illness [1]. Some authors define it as a subjective, complex and non-specific phenomenon occurring when the demands of a process or situation exceed the available resources and recovery mechanisms are not sufficient [2]. Tools such as the Medical Subject Headings (MeSH) thesaurus define it as ‘the state of weariness following a period of exertion, mental or physical, characterized by a decreased capacity for work and reduced efficiency to respond to stimuli’ [3] in an effort to standardize the scientific use of this term, combining its multidimensional and multicausal nature.

Fatigue is also a highly prevalent condition that has been studied in a large number of patient populations with pathologies such as cancer [4,5,6], heart attack [7,8,9,10,11], kidney disease [12] or sarcoidosis [13,14,15,16], but also during pregnancy and the postpartum period [17,18,19], as well as in work environments [20,21,22,23]. 

Nevertheless, there is little research in the field of palliative care caregivers. Symptoms of fatigue, loss of energy and tiredness are also very common in populations of family caregivers, especially those of palliative care patients. Among other issues, this is due to the high care demands of palliative care patients, as well as the complexity of end-of-life situations [24]. Caregivers of palliative care patients experience high levels of physical and emotional burden. In most cases, they are dedicated to the service of the person(s) they care for, at the expense of issues such as their social relationships, professional career or their own self-care [25,26,27,28]. 

Fatigue in caregivers of palliative care patients has been associated with variables such as caregiver burden and depression, as well as with other variables such as self-perception of health, symptoms, including anxiety or drowsiness, apathy and sociodemographic variables such as the age of the caregivers [18,27,29,30,31,32,33,34]. 

Despite the high impact of fatigue levels on their quality of life, few national or international studies have been found that specifically focus on the assessment and treatment of fatigue in caregivers of palliative care patients [27,30]. In terms of the clinical assessment of fatigue levels in this population, in the Spanish context there is currently no tool specifically validated for assessing fatigue in these caregivers. This makes it difficult to assess fatigue and develop prevention and care interventions.

The review by Rahimian Aghdam et al. [35] includes some of the main fatigue assessment tools that have been developed internationally over recent decades, including the Fatigue Assessment Scale (FAS). The FAS has the advantage of being a short, one-dimensional and easy to use tool that assesses both psychological and physical aspects. Since its publication [1], this scale has shown high internal consistency values and adequate evidence of validity with respect to related constructs, such as depression. It has also been used and validated for use with people with different pathologies such as heart attack or sarcoidosis, as well as in many different contexts and countries [10,13,14,15,36]. In Spain, the FAS was adapted and validated by Cano-Climent et al. [18] in a sample of postpartum women, but nothing is known of its psychometric properties (factor structure and evidence of validity and reliability) in other populations such as caregivers of palliative care patients.

The aim of the present study was to examine the FAS’s psychometric properties in family caregivers of terminally ill patients. The hypotheses were: (1) that the FAS will have a unifactorial structure, in line with the original version; (2) that it will have adequate reliability values greater than 0.70; (3) that the FAS total score will show moderate–high correlations with caregiver burden, anxiety and depression scores and low correlations with family functioning, life satisfaction, resilience and health-related quality of life scores (both physical and mental components); and (4) that significant differences between known groups (caregivers with and without substantial fatigue) will be evident in the family functioning, life satisfaction, resilience, caregiver burden, anxiety, depression and health-related quality of life variables.

## 2. Materials and Methods

### 2.1. Study Design 

An instrumental study aimed at generating or adapting an assessment tool, including an examination of its psychometric properties [37], was conducted in a previously unresearched population. Structural equivalence between the Spanish version and the original instrument was established and its internal consistency and construct validity were assessed.

### 2.2. Participants

Non-probability purposive sampling was used to select the sample, on the basis of the following inclusion criteria: (a) being a family caregiver of an adult patient who has an advanced chronic disease, is not cognitively impaired and requires palliative care; (b) being over 18 years of age; and (c) being able to understand and express themselves correctly in Spanish. Formal caregivers receiving financial compensation and those caring for patients with advanced dementia were excluded.

### 2.3. Variables and Instruments

An ad hoc data collection questionnaire was prepared which included the following standardized instruments:Fatigue Assessment Scale (FAS) [1].

This tool consists of 10 items assessing perceived physical and mental fatigue, with a 5-point Likert-type response scale ranging from 1 = “never” to 5 = “always”. Higher scores show higher levels of fatigue, with the cut-off point of 22 or more indicating substantial fatigue. The Spanish version (FAS-e) has a one-dimensional factor structure and adequate psychometric properties, with an internal consistency of 0.8 [18].

Family Adaptability, Partnership, Growth, Affection and Resolve (APGAR) [38].

This instrument was designed to assess individual perceptions of family functioning. It consists of 5 items, with Likert-type response alternatives ranging from 0 = “almost never” to 2 = “almost always”. Maximum punctuation is 10 and, the higher the score, the better the family functioning. It was adapted to Spanish by Bellón et al. [39], has a one-dimensional structure and adequate indicators of internal consistency and validity [39].

Satisfaction with Life Scale (SWLS) [40].

This questionnaire consists of 5 items assessing life satisfaction using a 7-point Likert-type response scale ranging from 1 = “Strongly Disagree” to 7 = “Strongly Agree”. Higher scores indicate higher life satisfaction. The Spanish version has a one-dimensional structure and adequate psychometric properties [41,42]. 

Caregiver Burden Interview (ZBI) [43]. 

This questionnaire was designed to assess caregiver burden and was adapted to Spanish by Martín et al. [44]. It is a multidimensional scale (physical, emotional-psychological, social and economic) of 22 items, with Likert-type response alternatives ranging from 0 = “never” to 5 = “almost always”. The score ranges from 22 to 110 points, and the higher the score, the higher the perceived caregiver burden. The scale is validated for different populations and has adequate psychometric properties, showing an internal consistency of 0.92 [45,46,47].

The Hospital Anxiety and Depression Scale (HADS) [48].

This questionnaire has 2 subscales of 7 items each, assessing anxiety and depression, respectively. The items have a 3-point Likert-type response scale, with higher scores indicating higher levels of anxiety and depression. It was adapted to Spanish by Tejero et al. [49] and has shown good validity and reliability indicators for both subscales [50].

Brief Resilient Coping Scale (BRCS) [51].

This instrument was developed to assess resilience in multiple populations and consists of 4 items assessed on a 5-point Likert-type scale ranging from 1 = “does not describe me at all” to 5 = “describes me very well”. Higher scores denote greater resilience. It was adapted to Spanish by Limonero et al. [52], showing good reliability and validity indicators.

12-Item Short Form Health Survey (SF-12) [53].

The SF-12 survey, a shortened version of the SF-36 Health Questionnaire by Alonso et al. [54], is a 12-item scale with eight dimensions (physical function, social function, physical role, emotional role, mental health, vitality, bodily pain and general health) which are divided into two components, physical and mental [55]. The items have different response alternatives depending on the dimension, assessing the intensity and/or frequency of the health condition. The initial score is transformed into a scale of 0–100, with a higher score implying a better health-related quality of life (HRQoL). It is a widely known and internationally used instrument with good psychometric validity and reliability indicators across different populations [55].

The following socio-demographic variables of the family caregiver were also included: age, sex, marital status, educational level and employment status, as well as the age, sex and condition of the patient being cared for. Three different models of the data collection questionnaire were prepared so that the presentation order of the different instruments would not affect subsequent questionnaires. Each version presented the instruments in a different order.

### 2.4. Procedure

Patients and their family caregivers were selected by the case management nurses in their primary care practices, who explained the aim of the study and performed an initial screening of those who wished to participate. A member of the research team contacted those who had expressed a wish to participate by telephone to arrange an appointment at their preferred location (their home or health center). On the agreed day, a researcher experienced in conducting interviews in the field of palliative care visited the home. Upon arrival, the study was presented and informed consent was obtained from both the patient and caregiver. The questionnaire was then given to the family member, who filled it in while the researcher interviewed the patient. At all times, both the patient and caregiver were afforded the privacy and freedom to answer questions as reliably as possible.

### 2.5. Ethical Considerations

This study is part of a research project by the University of Alicante, the Alicante Institute of Health and Biomedical Research (ISABIAL) and the University of Granada. The project was authorized by the Ethics and Health Research Committee of the Andalusian Autonomous Government and by the Foundation for the Promotion of Health and Biomedical Research (FISABIO) of Valencia Region (reference number: PI2017/66). During this work, all participants’ rights were respected and they gave their written informed consent. All personal data obtained in this study were processed in accordance with Organic Law 3/2018 of 5 December, Protection of Personal Data and Guarantee of Digital Rights, as established by Regulation (EU) 2016/679 of the European Parliament and of the Council of 27 April 2016 on Data Protection (GDPR).

### 2.6. Data Analysis

SPSS version 26.0 (IBM, New York, USA, USA) and R software (R Foundation, Vienna, Austria) were used. To test the factor structure, confirmatory factor analysis (CFA) was performed using the robust weighted least squares estimation (WLSMV) method of the Lavaan package in R, which is used for ordinal variables [56]. Three indices were considered to analyze the model fit for categorical variables [57]: the comparative fit index (CFI), the Tucker–Lewis Index (TLI), and root mean square error of approximation (RMSEA). For the CFI and TLI indices, adequate values are considered to be those greater than 0.90 [58]. For the RMSEA index, values between 0.05 and 0.08 indicate a reasonable fit [59]. The analysis of internal consistency was obtained using the non-linear reliability estimator based on the structural equation model (SEM) [59,60] suggested for ordinal data [61], and extracted using the semTools package. Bivariate Pearson correlations between the FAS total score and measures of family functioning, life satisfaction, caregiver burden, anxiety, depression, resilience and health-related quality of life were used for validity evidence. Values above ±0.7 were considered high correlations, values from ±0.4 to ±0.6 considered moderate and below ±0.4 considered low [62]. Student’s *t*-tests were used to calculate differences between those participants who had significant levels of fatigue (total scores above 22) and those who did not.

## 3. Results

### 3.1. Description of the Sample

A total of 108 caregivers participated. The majority were women (66.7%) with a mean age of 60.5 years (SD = 12.85) and 83.3% of the participants were married or co-habiting. In terms of employment status, the largest single groups were retired people (29.6%) and homemakers (25.9%), and 39.8% had a primary level of education (see Table 1). The participating caregivers looked after patients with a mean age of 73.6 years (SD = 13.80), approximately half of whom were women (50.9%) and most of whom had been diagnosed with cancer (59.3%).

### 3.2. Factorial Structure and Reliability of the FAS 

A one-dimensional model was analyzed, on the basis of the model proposed in the Spanish version [18]. The results of the CFA showed that for most of the statistics, the data fit the one-dimensional model. The exception was the RMSEA statistic, which was at the limit of acceptability for this index (0.088). Therefore, the covariance of the errors between items 4 and 10 proposed by the Spanish version [18], and as suggested by the analysis of the modification indices, was included in the one-dimensional model (see Table 2). Given the analysis of measures with correlated errors, the fit of the data to the structural model is adequate: CFI = 0.991; TLI = 0.988; RMSEA = 0.078. Factor loadings ranged from 0.72 to 0.87, except for two items with lower loadings: item 3 (λ = 0.23) and item 10 (λ = 0.41). The internal consistency calculated with the non-linear SEM reliability coefficient was 0.88 (see Table 2). 

### 3.3. Evidence of Validity 

Table 3 shows the mean scores of the different scales used to test the validity evidence of the FAS. It can be seen that the mean total score for caregiver fatigue was 23.31 (SD = 8.31). Statistically significant positive relationships were observed with the caregiver burden, anxiety and depression variables, as well as statistically significant negative associations with family functioning, life satisfaction, resilience and psychological quality of life (see Table 3).

When participants were divided according to their FAS score, there were 54 participants with substantial fatigue (scores above 22) and 51 participants without fatigue. Differences between these groups were found in terms of family functioning (*p* = 0.003), life satisfaction (*p* = 0.040), caregiver burden (*p* < 0.001), anxiety (*p* < 0.001), depression (*p* < 0.001) and the mental component of quality of life (*p* = 0.018) (See Table 4).

## 4. Discussion

The aim of this study was to provide psychometric evidence for the Spanish version of the FAS for use with family caregivers of palliative care patients. The findings of the study have confirmed previous hypotheses, showing that the scale has adequate psychometric properties. 

With respect to the first hypothesis, the confirmatory factor analysis showed adequate fit indices for the one-dimensional structure, in line with the original instrument [1,29], as well as its validity in different populations [17,18,63]. However, it is worth noting that certain versions of the FAS have a two-dimensional structure, distinguishing between both the mental and the physical components of fatigue [10,16,17]. To be specific, Giallo et al. [17] showed a good data fit for both the one-dimensional and two-dimensional structures. However, the unifactorial structure was recommended as more appropriate due to the principle of parsimony, high correlation between the factors and high factor loadings on the two-factor items.

With respect to the second hypothesis, the scale showed good internal consistency, in line with different versions of the FAS [14,18,20,36,63], with internal consistency ranging from 0.80 to 0.93.

With regard to hypotheses 3 and 4, concerning evidence of construct validity in caregivers of palliative care patients, it is confirmed that the scale shows statistically significant, positive and moderate–high relationships with caregiver burden levels and anxiety and depression scores, as well as statistically significant, negative and low relationships with family functioning, life satisfaction, resilience and psychological quality of life. Correlations of fatigue with caregiver burden and anxiety and depression have already been widely reported in the literature [1,18,33,64]. This evidence of convergent and divergent validity shows adequate construct validity with respect to other variables in the nomological network. Finally, the evidence of discriminant validity between the known groups (caregivers with and without substantial fatigue) also supports the construct validity of the FAS. Although most variables showed differences between the groups, no differences were obtained in terms of resilience levels. While resilience has been associated with a higher resistance to stress in caregivers of palliative care patients, its relationship with fatigue remains less clear. A recent review by Palacio et al. [65] showed that resilient coping in caregivers was positively associated with quality of life, mental health and personal growth, but included no information about its association with fatigue levels. Further studies are therefore required to study the relationships between the two variables. 

The study’s main limitations are the high number of women in the sample, together with the high number of cancer diagnoses, reflecting the profile of caregivers in Spain. Although it is not an excessively large sample, the recommendation of a minimum of 10 observations for each independent variable is met [66].

The present study has clinical implications in the current population. Fatigue seems to be a central variable that affects caregivers of PC patients and that is associated with family functioning, life satisfaction, caregiver burden, anxiety, depression and the mental component of quality of life. The assessment of fatigue in caregivers of palliative care patients using a validated tool will enable healthcare professionals to ascertain and specifically monitor their fatigue levels. It will also help in the evaluation of new interventions for preventing fatigue and improving the quality of life of these caregivers through an individualized and needs-based approach [27,33]. Further studies are needed using both quantitative and qualitative approaches to fatigue. For example, the use of Online Photovoice (OPV), a qualitative methodology that gives opportunities to the participants to express their own experience with little manipulation [67], may be useful for studying the experience of fatigue in caregivers of PC patients. 

## 5. Conclusions

In conclusion, the results of this study indicate that the Spanish version of the FAS scale in family caregivers of palliative patients has a one-dimensional structure, with adequate internal consistency values and satisfactory evidence of construct validity.

## Figures and Tables

**Table 1 jcm-11-03999-t001:** Sociodemographic data of participants (*n* = 108).

Variables	*n* (%)
**Sex**	
Male	36 (33.3)
Female	72 (66.7)
**Civil Status**	
Single	9 (8.3)
Married or living with a partner	90 (83.3)
Widowed, divorced or separated	9 (8.3)
**Employment**	
Part-time	8 (7.4)
Full-time	21 (19.4)
Unemployed	19 (17.6)
Retired	32 (29.6)
Homemaker	28 (25.9)
**Educational Level**	
No formal education	16 (14.8)
Primary	43 (39.8)
Secondary	34 (31.5)
University	15 (13.9)
**Patient’s sex**	
Male	53 (49.1)
Female	55 (50.9)
**Patient’s diagnosis**	
Cancer	64 (59.3)
Organ insufficiencies	28 (25.9)
Neurodegenerative disease	16 (14.8)

**Table 2 jcm-11-03999-t002:** Results of the confirmatory factor analysis (*n*= 108).

	Chi-Square	df	p	CFI	TLI	RMSEA	Internal Consistency
Mod 1	69.827	35	0.002	0.988	0.984	0.088	0.89
Mod 2	56.020	34	0.010	0.991	0.988	0.078	0.88

Note: Mod 1 = one-dimensional model; Mod 2= one-dimensional model with correlated errors between item #4 and item #1; CFI = comparative fit index; TLI = Tucker–Lewis Index; RMSA = root mean square error of approximation.

**Table 3 jcm-11-03999-t003:** Bivariate Pearson correlations between FAS and the variables of family functioning, life satisfaction, caregiver burden, anxiety, depression, resilience and health-related quality of life.

	Mean Score (SD)	Apgar	SWLS	Zarit	HADS ANX	HADS DEP	BRCS	SF-12 PH	SF-12 MH
FAS	23.31 (8.31)	−0.33 ***	−0.30 **	0.56 ***	0.63 ***	0.67 ***	−0.28 **	0.08	−0.28 **
Apgar	8.37 (2.06)	1	0.28 **	−0.37 ***	−0.22 *	−0.36 ***	0.32 **	−0.01	0.07
SWLS	17.19 (4.66)		1	−0.26 **	−0.27 **	−0.40 ***	0.25 **	−0.20 *	0.11
Zarit	52.20 (15.66)			1	0.62 ***	0.62 ***	−0.24 *	−0.07	−0.26 **
HADS ANX	9.97 (4.73)				1	0.73 ***	−0.26 **	0.01	−0.34 ***
HADS DEP	6.69 (4.36)					1	−0.32 **	0.12	−0.39 ***
BRCS	13.99 (3.24)						1	−0.12	0.12
SF-12 PH	41.49 (6.15)							1	−0.49 ***
SF-12 MH	40.64 (9.39)								1

Note. *** *p* < 0.001, ** *p* < 0.01, * *p* < 0.05. FAS = Fatigue Assessment Scale; SWLS = Satisfaction with Life Scale; HADS ANX = The Hospital Anxiety and Depression Scale–Anxiety Subscale; HADS DEP = The Hospital Anxiety and Depression Scale–Depression Subscale; BRCS = Brief Resilient Coping Scale; SF-12 PH = 12-Item Short Form Health Survey Physical Health; SF-12 MH = 12-Item Short Form Health Survey Mental Health.

**Table 4 jcm-11-03999-t004:** Differences between groups with substantial fatigue and without fatigue in family functioning, life satisfaction, caregiver burden, anxiety, depression and health-related quality of life.

	Group	Mean	SD	*t*	*p*
Apgar	Without Fatigue	8.96	1.54	3.04	0.003
	Substantial Fatigue	7.80	2.34		
SWLS	Without Fatigue	18.13	5.05	2.09	0.040
	Substantial Fatigue	16.29	4.10		
Zarit	Without Fatigue	45.51	12.50	−4.79	<0.001
	Substantial Fatigue	58.65	15.77		
HADS ANX	Without Fatigue	7.57	4.13	−5.98	<0.001
	Substantial Fatigue	12.29	4.08		
HADS DEP	Without Fatigue	4.21	3.10	−6.96	<0.001
	Substantial Fatigue	9.07	4.07		
BRCS	Without Fatigue	14.43	2.82	1.41	0.164
	Substantial Fatigue	13.56	3.58		
SF-12 PH	Without Fatigue	41.11	5.27	−0.62	0.537
	Substantial Fatigue	41.85	6.91		
SF-12 MH	Without Fatigue	42.85	8.15	2.40	0.018
	Substantial Fatigue	38.56	10.06		

Note. FAS = Fatigue Assessment Scale; SWLS = Satisfaction with Life Scale; HADS ANX = The Hospital Anxiety and Depression Scale–Anxiety Subscale; HADS DEP = The Hospital Anxiety and Depression Scale–Depression Subscale; BRCS = Brief Resilient Coping Scale; SF-12 PH = 12-Item Short Form Health Survey Physical Health; SF-12 MH = 12-Item Short Form Health Survey Mental Health.

## Data Availability

Not applicable.

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
