# Peer review of "Psychometric Properties of the Spanish Version of the Fatigue Assessment Scale in Caregivers of Palliative Care Patients"

_jcm, 2022, doi:10.3390/jcm11143999_

Round 1
Reviewer 1 Report
The study is relevant for the field and examines an important topic and population group. The Spanish version of the FAS could be useful for future research.
The manuscript is well presented.
But check:
1. the lines 107-108: „Likert scale… 2=almost always“ or it should be 5? Similar lines 112-113: Scale 1-5 or 1-7?
2. Table 2: “Consistencia interna” this has to be in English?
The study design is appropriate and the details given in the methods section are consistent and clear.
The results are easy to understand and the conclusions consistent with the presented evidence
Author Response
Dear Reviewer,
We want to thank your thoughtful reading, your effort, suggestions and quick response.
Please, find attached a document with the changes we made according to your suggestions.
Best regards,
The Authors

Reviewer 2 Report
Dear researcher(s), you are addressing an important and meaningful gap. Your paper is well-written and it has some important results, and if you edit your paper it can be much more effective. Here some humble suggestions to improve the paper, I would do the following to strengthen the paper. please see attached review.

Author Response

(The authors gave the same response as above.)
